# Insensitive High-Energy Density Materials Based on Azazole-Rich Rings: 1,2,4-Triazole *N*-Oxide Derivatives Containing Isomerized Nitro and Amino Groups

**DOI:** 10.3390/ijms24043918

**Published:** 2023-02-15

**Authors:** Xinbo Yang, Nan Li, Yuchuan Li, Siping Pang

**Affiliations:** 1School of Mechatronical Engineering, Beijing Institute of Technology, Beijing 100081, China; 2School of Materials Science and Engineering, Beijing Institute of Technology, Beijing 100081, China

**Keywords:** triazole derivative, electronic structure calculation, stability, detonation performance, insensitive energetic material

## Abstract

It is an arduous and meaningful challenge to design and develop new energetic materials with lower sensitivity and higher energy. How to skillfully combine the characteristics of low sensitivity and high energy is the key problem in designing new insensitive high-energy materials. Taking a triazole ring as a framework, a strategy of *N*-oxide derivatives containing isomerized nitro and amino groups was proposed to answer this question. Based on this strategy, some 1,2,4-triazole *N*-oxide derivatives (**NATNO**s) were designed and explored. The electronic structure calculation showed that the stable existence of these triazole derivatives was due to the intramolecular hydrogen bond and other interactions. The impact sensitivity and the dissociation enthalpy of trigger bonds directly indicated that some compounds could exist stably. The crystal densities of all **NATNO**s were larger than 1.80 g/cm^3^, which met the requirement of high-energetic materials for crystal density. Some **NATNO**s (9748 m/s for **NATNO**, 9841 m/s for **NATNO-1**, 9818 m/s for **NATNO-2**, 9906 m/s for **NATNO-3**, and 9592 m/s for **NATNO-4**) were potential high detonation velocity energy materials. These study results not only indicate that the **NATNO**s have relatively stable properties and excellent detonation properties but also prove that the strategy of nitro amino position isomerization coupled with *N*-oxide is an effective means to develop new energetic materials.

## 1. Introduction

The high-energy-density materials (HEDMs) bear numerous applications in aerospace and military fields [1]. In HEDMs, energy and sensitivity are always a pair of contradictions, which greatly restrict its application and development [2]. How to skillfully combine the characteristics of low sensitivity and high energy is the key problem in designing new insensitive high-energy materials.

Referring to two representative aromatic-insensitive high-energy materials, 1,1-diamino-2,2-dinitroethene (FOX-7) [3] and 2,4,6-triamino-1,3,5-trinitrobenzene (TATB), it can be found that they have similarities in structures: (1) a conjugated planar molecular skeleton is helpful for molecules to form a layered structure in a crystal structure with π–π packing, which can reduce the destructive effect of external stimulation [4], leading to their high stability, and (2) a nitro group with an electron-absorbing effect and an amino group with an electron-donating effect are adjacent to each other to form a strong intramolecular hydrogen bond [5], which further improves the molecular stability. So they are often used as the ideal insensitive explosives [6]. Instead, aliphatic nitramine energetic materials generally have higher energy, such as hexanitrohexaazaisowurtzitane (CL-20) [7], octahydro-1,3,5,7-tetranitro-1,3,5,7-tetrazocine (HMX) [8], hexahydro-1,3,5-trinitro-1,3,5-triazine (RDX) [9], 1,3,3-trinitroazetidine (TNAZ) [10], etc. There are many features in their molecules, such as ring structure, more nitro groups, and N–NO_2_ bonds. In addition, the nitrogen atom oxidation strategy (*N*-oxides) of heterocyclic compounds can also improve their energetic properties [11]. By combining the structural characteristics of aromatic energetic materials with better safety and aliphatic energetic materials with higher energy, it is possible to obtain new insensitive energetic materials with higher-energy performance and better safety.

As one of the aromatic energetic frameworks, azoles have attracted more and more attention in recent years due to their high nitrogen content, high density, and positive enthalpy of formation [12]. Among them, energetic materials with triazole as the building block are easy to further modify by various energetic groups due to the existence of C–H bonds to obtain energetic derivatives with rich structures. For example, the energetic properties of triazol-5-one-*N*-oxides and their energetic derivatives were studied in detail [13]. Combining nitro, amino, and 1,2,3-triazole structures, a high-performance energetic material with a detonation velocity and a detonation pressure of 8843 m/s and 36.2 GPa was synthesized [14]. By increasing the number of nitro groups and oxygen atoms, 3, 3′-dinitro-5, 5′-bis-1,2,4-triazole-1, 1′-diol, and their nitrogen-rich salts thereof were synthesized [15]. The decomposition temperatures of these compounds are located in the range of 207–329 °C, and the detonation velocities are located in the range of 8102–9087 m/s. Based on a polyamino-substituted strategy, the energetic properties and safety of azo-1,2,4-triazole can be significantly improved [16]. Using the fused-ring aminotriazole structure as a building block, 3,6,7-triamino-7*H*-[1,2,4]triazolo [4,3-b][1,2,4]triazole (TATOT) was designed and synthesized [17]. A simple oxidation synthesis method was used to achieve selective oxidation from 3,5-diamino-1,2,4-triazole to 3-amino-5-nitro-1,2,4-triazole (ANTA) [18]. Yan et al. [19] designed and synthesized a series of nitrogen-rich/oxygen-rich energetic materials, in which 5′-nitro-1,2′-bis(trinitromethyl)-1*H*, 2′*H*-3, 3′-bis(1,2,4-triazole) had high density (1.92 g/cm^3^), positive oxygen balance, and detonation performance comparable with that of HMX. In addition, using 2,4,6-triamino-5-nitropyrimidine-1, 3-dioxide hydrate as the raw material, an insensitive high-energy molecule closer to ideal, 3,5-diamino-6-hydroxy-2-oxide-4-nitropyrimidinone (IHEM-1), which included nitro, amino, and *N*-oxide [20], was synthesized [21]. Its detonation velocity and detonation pressure were 8660 m/s and 33.64 GPa, respectively, which were better than those of TATB. These strategies of amination, nitration, and oxidation on triazole really enrich the development of energetic materials. However, these studies appear to cause people to realize that the energy and safety of triazole energetic materials have reached the bottleneck period, and it is difficult to make a breakthrough.

In order to understand and design an excellent insensitive high-energy material, based on the 1,2,4-triazole skeleton, a strategy of coupling positional isomerized nitro and amino groups with *N*-oxides was proposed. Based on this strategy, seven 1,2,4-triazole *N*-oxide derivatives (**NATNO**s) containing nitro and amino substitution were designed, which included 3,5-diamino-1-nitro-1,2,4-triazole-2,4-dioxide (**NATNO**), 4-amino-3,5-dinitro-1,2,4-triazole-1-oxide (**NATNO-1**), 1-amino-3,5-dinitro-1,2,4-triazole-4-oxide (**NATNO-2**), 1-amino-3,5-dinitro-1,2,4-triazole-2-oxide (**NATNO-3**), 1,5-diamino-3-nitro-1,2,4-triazole-2,4-dioxide (**NATNO-4**), 2,4-diamino-3,5-dinitro-1,2,4-triazole-1-oxide (**NATNO-5**), and 1,2-diamino-3,5-dinitro-1,2,4-triazole-4-oxide (**NATNO-6**). The electronic structure properties of **NATNO**s were studied by molecular van der Waals surface electrostatic potential (ESP) analysis, interaction region indicator function (IRI) analysis, and anisotropy of current-induced density (ACID). The trends of the stability of **NATNO**s are discussed in terms of the bond dissociation enthalpy (BDE) of the trigger bond. Finally, the calculation of the detonation parameters shows that **NATNO**s have very excellent energetic properties, which further responds to the rationality of the design strategy proposed in this paper. Through the study of their electronic structure, stability, and detonation performance, the research results have a certain guiding significance for understanding the energetic properties and experimental synthesis of these substances.

## 2. Results and Discussion

### 2.1. Configuration

Figure 1 shows an optimized structure of the designed **NATNO**s (**NATNO**, **NATNO-1**, **NATNO-2**, **NATNO-3**, **NATNO-4**, **NATNO-5**, and **NATNO-6**) that were confirmed to have global minima without imaginary frequencies at the M06-2X/6-311G (d, p) level. NN bonds are important energy sources of energetic materials [22]. The bond length of a molecule reflects the stability of the bond to a certain extent [23]. Table 1 contains the bond length of each chemical bond in the **NATNO**, and other molecules’ chemical bonds can be found in Appendix A. In **NATNO,** the distances of N6–N10, N6–N7, N10–O11, and N15–O16 were 1.449 Å, 1.460 Å, 1.268 Å, and 1.294 Å, respectively. The Mayer bond orders of them were 0.856, 0.802, 1.251, and 1.161, respectively. The bond length and bond order suggested the existence of an NN single bond and an N=O bond in the molecule. In addition, the bond length was shorter, and the Mayer bond order was larger, suggesting that N6–N10 bond breaking was more likely to occur in **NATNO** molecules. On the whole, **NATNO**s had the structural characteristics of nitro and amino isomerization, a planar skeleton, and *N*-oxides. In addition, the nitro and triazole ring planes had a certain intersection angle (47.84°), which implied that the repulsion between nitro and oxygen atoms (O11) hindered the planar conjugation tendency between them. By swapping the amino and nitro positions in the **NATNO** molecule, the **NATNO-1** was formed. The N–NH_2_ (N6–N11) bond length in **NATNO-1** was 1.3687 Å, which was longer than that of N–NO_2_ (N6–N7, 1.460 Å) in **NATNO**.

In addition, comparing the bond length and bond order of the N–N bond in the 1,2,4-triazole ring of each **NATNO** molecule, it was found that the order of the bond lengths of the seven molecules was **NATNO** (Bond_N6–N10_, 1.449 Å) > **NATNO-6** (Bond_N17–N18_, 1.438 Å) > **NATNO-5** (Bond_N13–N15_, 1.421 Å) > **NATNO-1** (Bond_N6–N7_, 1.393 Å) > **NATNO-4** (Bond_N9–N14_, 1.386 Å) > **NATNO-2** (Bond_N4–N14_,1.353 Å) > **NATNO-3** (Bond_N3–N10_, 1.301 Å), and the corresponding bond order was **NATNO** (MBO_N6–N10_, 0.856) < **NATNO-6** (Bond_N17–N18_, 0.881) < **NATNO-5** (Bond_N13–N15_, 0.905) < **NATNO-1** (MBO_N6–N7_, 0.952) < **NATNO-4** (Bond_N9–N14_, 0.990) < **NATNO-3** (MBO_N3–N10_, 1.274) < **NATNO-2** (MBO_N4–N14_, 1.353), which implied that the stability order of the ring structure in these molecules may have been **NATNO** < **NATNO-6** < **NATNO-5** < **NATNO-1** < **NATNO-4** < **NATNO-3** < **NATNO-2** (see Table 2). The difference of the bond length and the bond order indicated that the position isomerism had a certain influence on the molecular structure.

### 2.2. Electrostatic Potential on Molecular Surface

The molecular van der Waals (vdW) surface electrostatic potential (ESP) [24,25,26] is an important tool to understand the noncovalent interactions and chemical reactions of energetic materials. Figure 2 represents the isosurface map of the molecular surface electrostatic potential of twelve kinds of molecules, the molecular structures of which were partially similar. The surface local minima and maxima of the ESP are represented as cyan and orange spheres (in kcal/mol). The surface minima of ESP basically appeared near the atoms with high electronegativity, such as oxygen and nitrogen atoms in nitro, while the surface maxima preferred to be mainly distributed near the atoms with low electronegativity, such as the hydrogen atom in the amino group and the carbon atom in the triazole ring. The global maxima of the ESP on ANTA, HDNT (3,5-dintiro-1,2,4-triazole), TATB, FOX-7, IHEM-1, **NATNO**, **NATNO-1**, **NATNO-2**, **NATNO-3**, **NATNO-4**, **NATNO-5**, and **NATNO-6** were 66.61, 74.04, 35.83, 68.01, 58.68, 51.62, 51.94, 60.70 58.15, 58.18, 53.88, and 69.76 kcal/mol, respectively, indicating that the order of the reactive activities of these nine compounds was HDNT > **NATNO-6** > FOX-7 > ANTA > **NATNO-2** > IHEM-1 > **NATNO-4** > **NATNO-3** > **NATNO**-**5** > **NATNO-1** > **NATNO** > TATB when attacked by nucleophiles. In addition, the position isomerization of the nitro and amino groups affected the extremum of the electrostatic potential on the molecular surface. Compared with **NATNO**, in **NATNO-1**, the distribution of oxygen atoms was more concentrated due to isomerization, which caused the extremum of the electrostatic potential in **NATNO-1** to appear larger. This isomerization caused the **NATNO-1** to be more energetic than **NATNO**. The same phenomenon was also reflected in **NATNO-2**, **NATNO-3**, and **NATNO-4**. The global maximum of ESP in **NATNO-3** was 60.70 kcal/mol, which was larger than that of **NATNO-2** (51.94 kcal/mol) or **NATNO-4** (58.15 kcal/mol)**.**

At the same instant, Figure 3 represents the surface area distribution of molecules in different electrostatic potential regions. The vdW surface electrostatic potential distributions of these molecules were similar in that the positive electrostatic potentials (red regions) were distributed near the H atoms with weak electronegativity, while the negative surface electrostatic potentials (blue regions) were distributed near the nitrogen and oxygen atoms with strong electronegativity. In ANTA, HDNT, and FOX-7 molecules, the overall area of the negative electrostatic potential region was larger than that of the positive electrostatic potential, suggesting that these molecules had larger reaction regions and interacted with electrophilic reagents. For IHEM-1 and **NATNO** molecules, the area sizes of the positive electrostatic potential region and the negative electrostatic potential region were relatively close, and their distributions were close to the normal distribution, suggesting that the electrostatic potential distributions of these two molecules were relatively uniform and may have had low reaction activity. The **NATNO-1** had a large surface area in the range of 20 to 60 kcal/mol, suggesting that this molecule had high reactivity. The **NATNO-2**, **NATNO-3**, and **NATNO-4**, which are isomers of each other, had different electrostatic potential extrema and distributions. **NATNO-6** had a wider range of electrostatic potential distribution than **NATNO-5**, and the more positive electrostatic potential extrema implied that **NATNO-6** was more prone to chemical reactions than **NATNO-5**. For TATB, although the area of the positive electrostatic potential region was larger than that of the negative electrostatic potential, the global maximum of ESP was the smallest among the twelve molecules, so its reaction activity was the lowest. In other words, the position isomerizations of nitro and amino groups affected the surface area distributions of molecules.

### 2.3. Interaction Region Indicator Function Analysis

In order to visually show the interactions existing in the designed molecules, the interaction region indicator function (IRI) [27] analysis was performed on **NATNO**s using the Multiwfn program. The IRI isosurface clearly showed the intramolecular van der Waals interaction, steric hindrance, and chemical bonds at the same time. Figure 4 represents the interactions existing in **NATNO**s and other molecules. Our recommended coloring method of ${\mathrm{sign}(\lambda }_{2})\rho$ on IRI isosurfaces is shown in Appendix A; the corresponding chemical explanations are also given. Among the twelve compounds, the IHEM-1 and TATB molecules had six-membered rings with appropriate spatial structure, which caused the hydrogen atom in the molecule and the adjacent oxygen atom to form a strong hydrogen bond interaction, while the remaining five-membered triazole ring had no such effect. The **NATNO**s designed in this paper had more vdW interactions or hydrogen bonds (H-bond) than those of ANTA and HDNT, which further proved that the designed **NATNO**s had better stability among these compounds. In particular, there were strong intramolecular hydrogen bond interactions (**NATNO-2** and **NATNO-5**) and van der Waals interactions (**NATNO-6**) in the molecules of **NATNO-2**, **NATNO-5**, and **NATNO-6**. For FOX-7, the small vinyl skeleton caused the amino and nitro groups to be in close contact, which led to the formation of strong intramolecular hydrogen bonds between the amino and nitro groups such as that of TATB. Abundant intramolecular interactions helped to promote the stability of the molecules themselves.

Moreover, for revealing the π interaction regions in those molecules, the IRI-π [27] analysis was calculated here. Figure 5 reveals that in all molecules, the conjugated π orbitals were outside the molecular plane. The bluer the region on the IRI-π isosurface, the greater the π electron density, suggesting a stronger π effect; the greener the π electron density was, the weaker the π effect was. We could find that in TATB, which had the largest IRI-π isosurface distribution range, and the isosurface on the carbon atom of benzene ring was blue, suggesting that there were the largest π electron interaction region and the strongest π interaction in the TATB molecule. The HDNT and ANTA had similar π interaction regions. However, due to the electronegativity difference between the nitro and amino groups, HDNT had a slightly larger π interaction region than ANTA. The FOX-7 had the smallest IRI-π isosurface, suggesting that the π interaction region in the molecule was the smallest, and the color of the isosurface was green, indicating that it had relatively weak π interaction. In addition, the IRI-π isosurface area of IHEM-1 was similar to that of TATB, and the isosurface area was larger than that of **NATNO**s, indicating that its π electron density was higher, the π interaction was stronger, and the conjugation was better. Among **NATNO**s, we could clearly see that in the molecules of **NATNO**, **NATNO-5**, and **NATNO-6**, the π electron density on the triazole ring was significantly lower than that of **NATNO-1**, **NATNO-2**, **NATNO-3**, and **NATNO-4**, which suggested that the π interaction of **NATNO**, **NATNO-5**, and **NATNO-6** molecules was significantly lower than that of other molecules. However, IRI analyses of the isosurfaces of **NATNO-5** and **NATNO-6** showed that there were intramolecular hydrogen bonds and weak interactions in the molecules **NATNO-5** and **NATNO-6**, which helped to improve the stability of the molecules.

### 2.4. Anisotropy of Current-Induced Density (ACID)

In order to visualize the degree of electron delocalization in **NATNO**s, the anisotropy of current-induced density (ACID) [28] was used to visually analyze the designed molecules, which generated a molecular ring current induced under a given external magnetic field. The stronger the ring current, the stronger the aromaticity. The ACID isosurfaces of **NATNO**s for all π electrons are plotted in Figure 6. In **NATNO**, it was seen that the current-induced density of the closed ring on the triazole was poor, indicating that the aromaticity of this structure was very weak. Such an aromatic-destroyed triazole structure tended to have poor stability, while in **NATNO-1**, besides the ring current density on the two amino groups, the ring current on the nitro and *N*–O bonds and the ring current on the triazole formed a closed loop, suggesting that the structure as a whole had a stronger aromatic than that of **NATNO**. When one amino group in **NATNO** or **NATNO-1** was replaced with a nitro group, **NATNO-2**, **NATNO-3**, and **NATNO-4** were derived. It could be seen from the ACID isosurfaces of these three molecules that when the two H atoms in the amino group were on the same side of the triazole ring, the amino group could strengthen the closed ring current of the whole molecule (see **NATNO-2** and **NATNO-3**). However, when the two H atoms in the amino group were located on both sides of the triazole ring (see **NATNO-4**), the amino group and the triazole ring could not form a closed ring current, which indicated that the overall aromaticity of the molecule was poor. In addition, the ACID isosurfaces of **NATNO-5** and **NATNO-6** showed that the effect of substituents on their respective triazole rings resulted in the inability to form a whole closed ring current and reduced their aromaticity. The poor aromaticity was consistent with the results of IRI-π analysis above, which confirmed the triazole ring of **NATNO-5** and **NATNO-6** again. To sum up, in 1,2,4-triazole, the substitution effects of the nitro group and amino group had different effects on the aromaticity of the whole molecule, which also showed that these two energetic groups had certain effects on the stability of materials from another perspective.

### 2.5. Sensitivity and Stability

In order to evaluate the impact sensitivity and safety of **NATNO**s molecule, the impact sensitivity and bond dissociation enthalpy were explored.

#### 2.5.1. Impact Sensitivity

The sensitivity of the explosive represents the degree to which the energetic material ignites under the stimulus of external conditions. In this paper, the theoretical impact sensitivities (*h*_50_ [29]) of **NATNO**s were calculated to initially understand the basic properties of these energetic materials (see Table 3). The results showed that the *h*_50_ values of **NATNO**, **NATNO-1**, **NATNO-2**, **NATNO-3**, **NATNO-4**, **NATNO-5**, and **NATNO-6** were 55.68, 52.30, 42.63, 41.53, 30.94, 44.74, and 29.69 cm, respectively, which fully showed that the designed molecular structures had better stability than those of HDNT (26.09 cm) and CL-20 (12.95 cm). The *h*_50_ values of **NATNO** (55.68 cm) and **NATNO-1** (52.30 cm) were higher than that of IHEM-1 (50.78 cm). However, there are many methods [30] to calculate *h*_50_, and the accuracy of the prediction results varies. Therefore, we only preliminarily evaluated the impact sensitivity here.

#### 2.5.2. Bond Dissociation Enthalpy

Bond dissociation enthalpy (BDE) was used to describe the thermal stability of energetic compounds. It is generally believed that compounds containing nitro and amino groups will break from C–NO_2_, N–NO_2_, and N–NH_2_ when chemical reactions occur. The higher the value of BDE, the higher the energy required to destroy the chemical bond. In order to elucidate the thermodynamic stability of **NATNO** molecules, the BDE and Mayer bond order (MBO) [33] of possible trigger bonds were studied in this part.

Table 4 contains the BDE and MBO values of the relevant chemical bonds calculated for each molecule. The results showed that in **NATNO,** the BDE of **the** trigger bond (N–NO_2_) was 232.38 kJ/mol; in **NATNO-1**, the BDE of the trigger bond (N–NH_2_ (N6–N11)) was the lowest, which was 111.37 kJ/mol; in **NATNO-2**, the BDE of the trigger bond (N–NH_2_ (N3–N8)) was the lowest, which was 181.42 kJ/mol; in **NATNO-3**, the BDE of the trigger bond (C–NO_2_ (C1–N7)) was the lowest, which was 201.34 kJ/mol; and in **NATNO-4**, the BDE of the trigger bond (N–NH_2_ (N9–N10)) was the lowest, which was 204.27 kJ/mol. Encouragingly, the BDE values of the trigger bonds of these five compounds (except **NATNO-5** and **NATNO-5**) were larger than 84 kJ/mol [34], which met the stability requirement for a practical energetic material. In addition, according to the BDEs of the trigger bonds, the order of the thermodynamic stability of each substance in Table 4 was IHEM-1 (313.36 kJ/mol) > TATB (305.40 kJ/mol) > ANTA (299.16 kJ/mol) > FOX-7 (298.59 kJ/mol) > NTO (292.36 kJ/mol) > HDNT (286.00 kJ/mol) > **NATNO** (232.38 kJ/mol) > **NATNO-4** (204.27 kJ/mol) > **NATNO-3** (201.34 kJ/mol) > RDX (182.48 kJ/mol) > **NATNO-2** (181.42 kJ/mol) > HMX (179.86 kJ/mol) > CL-20 (171.96 kJ/mol) > **NATNO-1** (111.37 kJ/mol). Furthermore, the dissociation enthalpies of **NATNO-5** and **NATNO-6** were very small (−0.74 kJ/mol and 8.9 kJ/mol), suggesting that it was difficult for these two compounds to exist stably in theory. The appropriate BDE of the trigger bonds directly proved that some compounds designed in this paper had sufficient stability.

More interestingly, the position isomerism had a certain effect on the BDE of the trigger bonds of the NATNO molecule. For example, the molecules of **NATNO** and **NATNO-1** were isomers of each other. The isomerism of nitro and amino positions caused not only the molecular structure but also the molecular stability to be different (the BDE of the trigger bond of **NATNO-1** was almost 50% lower than that of **NATNO**). Second, although **NATNO-2**, **NATNO-3**, and **NATNO-4** were isomers of each other, the BDE of N–NH_2_ was lower than that of C–NO_2_ in **NATNO-2** and **NATNO-4**, while the situation of **NATNO-3** was the opposite (see Appendix A). This implied that the concentrated distribution of the *N*-oxides and the two nitro groups would lead to the enhancement of the activity of the directly adjacent nitrogen atoms in 1,2,4-triazole, thus making the region prone to chemical bond breakage. Similarly, in **NATNO-6**, the mode of unstable dissociation of the molecule was the breaking process of the chemical bond between the two adjacent nitrogen atoms in 1,2,4-triazole, resulting in the molecular ring opening.

### 2.6. Detonation Performance

The detonation performance represents the energy properties of energetic materials. Table 5 shows the detonation parameters of the compound **NATNO**s and other compounds. The oxygen balance of the **NATNO**s can be found in Table 4. Among the **NATNO**s, the isomers of **NATNO-2**, **NATNO-3**, and **NATNO-4** belonged to zero oxygen balance in molecular structure, suggesting that the full combustion of these molecules would produce the largest number of gaseous products and thus have better detonation performance.

The enthalpies of formation of all **NATNO**s (400.6 kJ/mol for **NATNO**, 460.3 kJ/mol for **NATNO-1**, 486.6 kJ/mol for **NATNO-2**, 505.4 kJ/mol for **NATNO-3**, 469.0 kJ/mol for **NATNO-4**, 673.4 kJ/mol for **NATNO-5**, and 751.8 kJ/mol for **NATNO-6**) were higher than these of others. The high and positive enthalpies of solid formation ensured that these molecules had high energies. In addition, the crystal densities of energetic materials were closely related to their detonation velocities and detonation pressures. The crystal densities of all **NATNO**s (1.92 g/cm^3^ for **NATNO**, 1.92 g/cm^3^ for **NATNO-1**, 1.92 g/cm^3^ for **NATNO-2**, 1.94 g/cm^3^ for **NATNO-3**, 1.88 g/cm^3^ for **NATNO-4**, 1.87 g/cm^3^ for **NATNO-5**, and 1.92 g/cm^3^ for **NATNO-6**) were larger than 1.80 g/cm^3^, which met the requirement of high-energetic materials for crystal density.

The **NATNO**s in this article had excellent detonation performance. The calculated detonation velocities and detonation pressures of some **NATNO**s (9748 m/s, 43.11 GPa for **NATNO**; 9841 m/s, 44.42 GPa for **NATNO-1**; 9818 m/s,43.74 GPa for **NATNO-2**; 9906 m/s, 44.83 GPa for **NATNO-3**; and 9592 m/s, 40.98 GPa for **NATNO-4**, respectively) were comparable with that of CL-20 (9530 m/s, 43.30 GPa). In addition, although **NATNO-5** and **NATNO-6** had excellent detonation performance, their stability was poor, and they could not exist stably in general. As a result of the calculation results, it is a reasonable strategy to design energetic molecules with the structural features of isomerized nitro and amino groups and *N*-oxides.

## 3. Materials and Methods

Electronic structure calculations were carried out using the Gaussian 09 [49], ORCA [50], Multiwfn [51], and Shermo [52] program packages. The geometries of each structure corresponding to the stationary point on the potential energy surface (PES) of the specie studied were fully optimized using density functional theory at the M06-2X/6-311G (d, p) level [53,54]. The zero-point energies and thermal corrections to enthalpy (the correction factor is 0.97) and Gibbs free energy were computed at the same DFT level. The single-point electronic energies were afterward refined using PWPB95 [55] functional in conjunction with a def2-QZVPP [56] basis set.

To better reflect the effects of intermolecular interactions in the crystals, the crystal density was estimated using the improved equation shown as follows [57]:(1)$$\rho =\alpha (\frac{M}{V(0.001)}{)+\beta (\upsilon \sigma }_{\mathrm{tot}}^{2})+\gamma$$
where ${\upsilon \sigma }_{\mathrm{tot}}^{2}$ is derived from the molecular electrostatic potential calculation, and *α*, *β*, and *γ* were coefficients assigned through fitting to the experimental densities of a series of 36 energetic compounds.

Enthalpy of formation ($∆H_{f}$) [58] is one of the most important parameters for energetic compounds. The definition of the enthalpy of formation (Figure 1) is used to calculate the gaseous enthalpy of formation of a substance as Equation (2):
(2)$$∆H_{f}\left(\mathrm{solid},\;298\;K\right)=∆H_{\mathrm{sub}}\left(\mathrm{graphite},\;298\;K\right)+∆H_{2}-∆H_{1}$$

The enthalpy of sublimation either vaporization [59] can be represented as Equation (3):(3)$$∆H_{1}{=a(\mathrm{SA})}^{2}+b\sqrt{\sigma_{\mathrm{tot}}^{2}\upsilon }+c$$
where SA is the molecular surface area for compound structure,$\;\sigma_{tot}^{2}$ is described as an indicator of the variability of the electrostatic potential on the molecular surface, and *υ* is interpreted as showing the degree of balance between the positive and negative potentials on the molecular surface where a, b, and c are fitting parameters. It is noteworthy that some recent work [60,61] has been conducted to obtain the crystal density based on crystal phase optimization and to estimate the enthalpy of formation based on the chemical molecular formula. However, we are not familiar with first-principles calculation at present, so we chose quantum chemical calculation.

The bond dissociation enthalpy (BDE) of the trigger bond is an important descriptor that can be used to describe the thermal stability of energetic materials.
(4)$${\mathrm{BDE}}_{\left(AB\right)}=\left[E_{A^{.}}+E_{B^{.}}\right]-E_{AB}$$
where *BDE*_(*AB*)_ represents the bond dissociation enthalpy of AB, $E_{A^{.}}$($E_{B^{.}}$) refer to the enthalpy of free radical $A^{.}$($B^{.}$), and *E_AB_* is the enthalpy of compound *AB*.

The detonation velocity (*D*) and pressure (*P*) were predicted by the empirical Kamlet–Jacobs [62] equation, which was carried out using the EXPLO5 (v6.05) [63] program:(5)$$D\;=1.01{\left(N{\overset{-}{M}}^{\frac{1}{2}}Q^{\frac{1}{2}}\right)}^{\frac{1}{2}}(1+1.30\rho )$$
(6)$$P\;=1.558\rho^{2}N{\overset{-}{M}}^{\frac{1}{2}}Q^{\frac{1}{2}}$$
where *D* is the detonation velocity (km/s), *P* is the detonation pressure (GPa), *N* is the moles of detonation gases per gram explosive, $\overset{-}{M}$ is the average molecular weight of these gases, *Q* is the heat of detonation (cal/g), and *ρ* is the loaded density of explosives (g/cm^3^) and is replaced by the theoretical density here.

The sensitivity of the energetic materials describes the difficulty of the explosive reaction of the energetic materials under an external stimulation. In this paper, only the impact sensitivity ($h_{50}$) [29,64] is evaluated by Equation (7).
(7)$$h_{50}=-0.0064\sigma_{+}^{2}+241.42\upsilon \;-\;3.43$$
where $h_{50}$ is the impact sensitivity, $\sigma_{+}^{2}$ ((kcal mol^−1^)^2^) is the indicator of the strengths and variabilities of the positive surface potentials, and *υ* is an electrostatic balance parameter.

## 4. Conclusions

In summary, taking 1,2,4-triazole as the molecular skeleton, based on the strategy of coupling positional isomerized nitro and amino groups with *N*-oxides, some 1,2,4-triazole derivatives (**NATNO**s) with excellent energy performance and moderate thermal stability were designed and explored. The electrostatic potential (ESP) analysis, the interaction region indicator function (IRI) analysis, and the analyses of the anisotropy of current-induced density (ACID) were fully performed to understand the properties of **NATNO**s. The calculation of impact sensitivity showed that **NATNO**s had moderate stability against external impact. Furthermore, the BDE values of the trigger bonds of these **NATNO**s were larger than 84 kJ/mol, which ensured the stable existence of some compounds in theory. The high density (>1.80 g/cm^3^) and positive enthalpy of formation (>377.4 kJ/mol) of **NATNO**s determined that these compounds had excellent detonation properties, the comprehensive detonation performance of which was completely comparable with that of CL-20. The strategy of coupling positional isomerized nitro and amino groups with *N*-oxides is worthy of application in the development of other new energetic compounds. We are further exploring relevant experimental synthesis and hope that other experts can jointly carry out relevant research works.

## Data Availability

Data can be found in the Appendix A.

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
