# Peer review of "Insensitive High-Energy Density Materials Based on Azazole-Rich Rings: 1,2,4-Triazole N-Oxide Derivatives Containing Isomerized Nitro and Amino Groups"

_ijms, 2023, doi:10.3390/ijms24043918_

Round 1

Reviewer 1 Report

This work presents an interesting theoretical study on some 1, 2, 4-triazole N-oxide derivatives (NATNOs) designed and tested for their impact sensitivities, energetic properties, and stabilities. A number of computational programs were employed, including Gaussian 09, ORCA, Multiwfn, and Shermo. The authors provide a thorough discussion of the properties of these compounds and found that in general they exhibit moderate stability against external impact and excellent detonation properties. The study seems to show that coupling positional isomerized nitro and amino groups with N-oxides results in the excellent energy performance and the improved stability over other explosive compounds. 

The study seems to be well designed, thought out, and researched. It is based on sound scientific principles and the conclusions seem appropriate for the data and results presented.  

I suggest publication after the following edits, clarifications, and improvements are made.

(1) I would like to see the "skeletal" molecular structures added to the supplementary materials for reference. I believe that this would help with visualizing the geometric structures provided in the main body of the paper.

(2) It is not necessary to say "The Figure 1" or "The Figure 2", etc. in the body of the text. You can simply say "Figure 1 shows..." or "Figure 1 represents...", etc.

(3)  There are many minor grammatical errors throughout the body of the text that will need to be corrected. 

(4) In line 248, delete "maybe". The results show better stability for the NATNOs. Using "maybe" in the sentence makes it seem as if you are not so sure.

(5) In lines 251-252, the sentence needs to be clarified. The use of "can make" does not make sense here. Perhaps replace them with "puts".

(6) In lines 157-160, the sentence starting with "The main molecular skeleton" is confusing as written and should be re-written for clarification.

(7) In line 300, what is meant by "open circuit"? Does this mean that these molecules have decreased conjugation? This should probably be written to drop that term.

(8) The formatting in Table 3 seems confusing. For some of the compounds, there are trigger bonds and data both above and below the compound listing. Is there a better way to line up the data with the compounds?

(9) In the paragraph that follows Table 3, would it be possible to add some speculation as to why similar bonds, such as Cl-NO2 bonds for example, for the different NATNOs vary from structure to structure as much as they do? This could be an interesting comparison for readers.

(10) In line 360, what is meant by "zero oxygen balance"? That term is not clear to this reviewer. 

Reviewer 2 Report

The authors present a theoretical study of a new insensitive energetic material. The work is interesting and worth to be published. But some methodological and stylistic issues must be responded first. Thus, a major revision is needed. My specific comments are the following:

1.       The authors used Scheme 1 and equations 1-3 to calculate crystalline density and solid-state enthalpy of formation via the calculations of vacuum-isolated molecular systems. This method is very popular although it is often a source of significant errors for different crystal systems. This is primarily due to inaccuracies in the estimation of the crystal densities. Recently, a relatively cheap and fast method for estimation of crystalline density and the solid-state enthalpy of formation was reported. I suggest the authors to consider this method. And, if not used in this work, then at least mentioned in the text as an alternative approach (https://doi.org/10.1016/j.fpc.2021.05.001, https://doi.org/10.1016/j.fpc.2021.09.002).

2.       The authors performed geometry optimizations and subsequent single-point energy corrections with different functionals and basis sets. At the same time, they used empirical equations 1-3, in which post-SCF calculations based on the electron density distribution analysis (Multiwfn) was also applied. It seems, the equations 1-3 are applicable for some “standard” level of theory. Meanwhile, the results of the Multiwfn calculations (and therefore the results of equations 1-3) may vary significantly. Did the authors consider such possibility and if yes, how this was taken into account?

3.       The authors state that thy are using “empirical Kamlet-Jacobs [33] with EXPLO5 (v6.05) [34] program”. This confuses me. As I know, EXPLO program uses the method, which on based on the iterative fitting the BKW or Murnaghan equation of states, rather than the afore-mentioned empirical scheme of K-J. Some clarification on this issue are necessary.

4.       Impact sensitivity estimation using equation 7 causes some concern for me. I suggest the authors to check its reliability of 1-2 related compounds with the experimentally known impact sensitivities.

5.       The paragraph (lines 171-183) is hard to read. I suggest rewriting it either in the form of a table (scheme) or in a more convenient form for the readers.

Round 2

Reviewer 2 Report

The authors responded to all my comments and I can see that the manuscript has become more rigor and clarity. Therefore, I can recommend it for publication in the present form.